# Optimization of the Use of a Commercial Phage-Based Product as a Control Strategy of *Listeria monocytogenes* in the Fresh-Cut Industry

**DOI:** 10.3390/foods12173171

**Published:** 2023-08-23

**Authors:** Marisa Gómez-Galindo, Pilar Truchado, Ana Allende, Maria I. Gil

**Affiliations:** Research Group on Microbiology and Quality of Fruits and Vegetables, Food Science & Technology Department, CEBAS-CSIC, 30100 Murcia, Spain; migomez@cebas.csic.es (M.G.-G.); ptruchado@cebas.csic.es (P.T.); aallende@cebas.csic.es (A.A.)

**Keywords:** food safety, fresh produce, biocontrol, foodborne, phage application

## Abstract

A commercial phage biocontrol for reducing *Listeria monocytogenes* has been described as an effective tool for improving fresh produce safety. Critical challenges in the phage application must be overcome for the industrial application. The validation studies were performed in two processing lines of two industry collaborators in Spain and Denmark, using shredded iceberg lettuce as the ready-to-eat (RTE), high process volume product. The biocontrol treatment optimized in lab-scale trials for the application of PhageGuard Listex^TM^ was confirmed in industrial settings by four tests, two in Spain and two in Denmark. Results showed that the method of application that included the device and the processing operation step was appropriate for the proper application. The proper dose of Phage Guard Listex^TM^ was reached in shredded iceberg lettuce and the surface was adequately covered for the successful application of phages. There was no impact on the headspace gas composition (CO_2_ and O_2_ levels), nor on the color when untreated and treated samples were compared. The post-process treatment with PhageGuard Listex^TM^ did not cause any detrimental impact on the sensory quality, including flavor, texture, browning, spoilage, and visual appearance over the shelf-life as the phage solution was applied as a fine, mist solution.

## 1. Introduction

*Listeria monocytogenes* (Lm) is a foodborne pathogenic bacterium that causes a spectrum of human illnesses (listeriosis) of variable severity [1]. Lm outbreaks have usually been linked to food of animal origin (FoAO) such as soft cheese or ready-to-eat (RTE) meats [2]. Given the recent Lm outbreaks in fruits and vegetables, there is a growing concern about Lm contamination in fresh produce and the risk to public health, which also includes the RTE fruit and vegetable industry [3].

Environmental monitoring programs established by the industry or carried out by research groups have shown that *Listeria* spp. can be isolated from fresh produce and processing environments [4,5]. The risk of Lm final produce contamination occurring via the industry environment is relatively low because of current safety management strategies implemented by the industry. The severity of the listeriosis outbreaks highlights the importance of effective preventive control strategies to reduce, control, and/or eliminate Lm [3]. Environmental monitoring programs and cleaning and sanitation plans implemented by the industry are in constant revision and improvement because of the absence of infallibility [6]. However, there are limits to the effectiveness of these measures that could lead to bacteria persistence in postharvest environments [6,7,8,9], plus the difficulties in having a representative microbial sampling for Lm detection.

The implementation of different post-process treatments as an additional safety barrier in fresh produce manufacturing has been extensively studied in several food products such as meat and dairy products, although very little in fresh whole products and almost unknown in RTE products of vegetal origin [10]. Increasing interest has been achieved in the post-process application of bacteriophages as natural pathogen-targeting bio-control agents to preserve food safety [11,12]. Phages are microorganisms of great abundance and ubiquity. These entities are involved in the dynamics of microbial populations in most ecosystems and have been used as antimicrobial agents for a century, but antibiotic use overshadowed their development [13]. In some countries, the use of different phages as biocontrol agents has been approved [14], although there are still some concerns regarding to their approval as processing aids. These concerns limit or delay the research in this area by hindering its application in commercial industrial environments and restricting its research studies to lab-scale trials [15,16]. In this sense, it is important to make a step forward in implementing research studies at the industrial level to validate Lm control strategies. This more applied research approach will help to gather information on phage optimization and effectiveness to gain industry confidence for the application of biopreservation treatments [15].

This study aimed to define the method and parameters of the application of a commercial phage-based treatment to control Lm in leafy greens at an industrial scale. The application of the previously optimized conditions of selected phage-based treatment had the goal to evidence under an industrial setting, the challenges for the application in the industry as well as its impact on the quality of the final product. The main objective of the validation trials was to demonstrate that the target concentration of the microorganisms needed to inhibit the growth of Lm was achieved under industrial conditions by adding the lowest amount of water to prevent any deterioration from the water excess on the product.

## 2. Materials and Methods

### 2.1. Biopreservative Agent and Application Conditions

PhageGuard Listex™ was the commercial biopreservative agent that contained a Phage P100 at a 10^11^ pfu/mL (Micreos, Wageningen, The Netherlands). The target concentration of the selected post-process treatment was 10^6^–10^7^ pfu/g. It was declared by the manufacturer to be effective against all *Listeria* strains. It is a USDA/FDA GRAS-approved (GRAS Notice No. 000218) and an FSIS processing aid approved when applied at a level of 10^7^ to 10^9^ pfu/g of the product (FSIS Directive 7120.1). It is further accepted as a processing aid in Australia, New Zealand, Israel, Switzerland, The Netherlands, Canada, and other countries.

Optimization of the utilization of PhageGuard Listex™ in an industrial setting was performed in two processing plants: one in Torre Pacheco (Murcia, Spain) and the other in Central Jutland Region (Arhus, Denmark). The fresh-cut product selected and processing lines in both locations were shredded iceberg lettuce. The application point at the Spanish facility was the vibration conveyor belt after the pre-visual inspection control point before ascending the conveyor belt to the packaging operation. At the Danish facility, the post-process treatment was applied in the vibrating conveyor belt just before the packaging machine entry. The two application points allowed the homogeneous mixture of the treated product before packaging.

For the application of the treatment in the processing plants, a prototype was built in an arc design above the selected conveyor belt with several nozzles to cover the conveyor width. The prototype consisted of a tank with the phage solution connected to a suction pump, which feed the nozzles that applied the treatment to the product. Nozzles were installed using a metallic structure placed above the conveyor belt that allowed the adjustment of the height of the nozzles. The same prototype was used in the two industrial settings. The post-process treatment was applied after washing and drying, just before packaging (Figure 1). For packaging, bags of RTE iceberg lettuce were flushed with nitrogen (N_2_) to reduce the oxygen (O_2_) concentration from 21% to 0–3%. Samples were analyzed after processing (day 0) and during the shelf-life (1, 5, 9 and 15 days of storage at 7 °C). However, the length of the shelf-life was adjusted depending on the product quality. Three out of four trials were cancelled after 9 days of storage because of the deterioration of the product, while in the other assay, a shelf life of 15 days was reached. The short shelf life was expected, as bags of 500 g of shredded iceberg were delivered for food service with a shelf life of a max of 6 days.

### 2.2. Microbiological Analyses

Bacteriophage enumeration was performed as described by Truchado et al., 2020 [11], following the spot test [17]. Briefly, a homogenate was obtained with twenty-five grams of the product and 225 of 0.1% sterile peptone water (SPW, wt/vol) (Scharlab). Polyethersulfone syringe filters of 0.45 μM (Thermo Fisher Scientific, San Jose, CA, USA) were used to obtain 1 mL of this homogenized without the presence of bacteria. Ten-fold serial dilutions were performed in SM buffer (5.8 g NaCl, 2.0 g MgSO_4_ 7H_2_O, 50 mL1 M Tris-HCl pH 7.4, 2% gelatin in 1 L dH_2_O) [18]. Three aliquots of this solution were placed on a lawn of the host strain. The plates were incubated at 30 °C 48 h. Detection of *Listeria* spp. and Lm in iceberg lettuce was based on the ISO-11290-1 with slight modifications. For enrichment, twenty-five grams of iceberg lettuce were mixed with 225 mL of Half Fraser Broth and 1% pyruvate and incubated for 48 h at 30 °C. Then, 1 mL of the homogenate was transferred to 9 mL of Fraser broth and incubated at 37 °C for 24 h. For enumeration, a part of the homogenized product in Half Frasher Broth was filtered through cellulose nitrate membrane filters (0.45 μM diameter, Sartorius, Madrid, Spain). The volume filtered of each sample varied between 1, 10 and 100 mL. Then, filters were placed in agar plates of ALOA/OCLA (Scharlau, Barcelona, Spain) [11].

Potentially positive colonies for Lm were isolated, grown in BHI (Oxoid, Thermo Fisher Scientific, San Jose, CA, USA) agar plates and tested by conventional polymerase chain reaction (PCR) using a PCR System (Applied Biosystems^®^ thermal cycler, Thermo Fisher Scientific, San Jose, CA, USA) with specific primers to confirm the presence of virulence *hly* and *iap* genes [19].

### 2.3. Sensory Evaluation

Commercial bags (n = 40) obtained in the industrial validation were transported to the laboratory (40 km) and kept at refrigerated storage conditions of 7 °C for 15 days.

Changes in the headspace gas composition (O_2_ and CO_2_ concentrations) within the bags were monitored each day of evaluation with a calibrated syringe and measured using a gas chromatograph (Shimadzu GC-14, Kyoto, Japan) equipped with a thermal conductivity detector (TCD). The gas was drawn from the bags using a septum attached to the bags.

The organoleptic attributes included browning, texture, flavor, spoilage, and overall visual quality of the control and treated product evaluated initially and after 5 and 9 days of storage. Coded (3 digits) samples were presented individually to six trained judges to make independent evaluations. Off-flavor and odor, texture, and spoilage were scored on a continuous scale ranging from 0 to 10 (0 = absence; 10 = completely damaged). Overall visual quality was scored on a continuous 0–10 scale (0 = extremely unpleasant; 10 = extremely pleasant).

At each sampling point, photographs (n = 4) of control and treated products were taken for the objective measurements of color changes [20]. The camera was a Canon EOS (70D) (Canon Lens: EF-S18-55mm f/3,5-5.6 IS, Canon Europe, Amstelveen, The Netherlands). Photograph conditions were: 1/50 s II shutter speed, ƒ/5,6 ISO: 100 aperture and SpyderCheckr™ RGB spectrum (v 1.3; Datacolor; Electrical & Electronic Manufacturing Lawrenceville, Trenton, NJ, USA) as a reference color chart for calibration. Photographs were processed (background removed and format conversion) with Adobe^®^ Photoshop^®^ 2020 (V 21.1.0; Adobe Photoshop CS. (2004). Peachpit Press: Berkeley, CA, USA). ImageJ (v 1.53 K; Rasband, W.S., ImageJ, U.S. National Institutes of Health, Bethesda, Maryland, USA. URL https://imagej.net/ij/; accessed on: 20 July 2023). RGB images were converted to Lab stack and the image values in the CIE L* a* b* color scale were obtained. In detail, L* indicates the lightness from black (0 value) to white (100 value), a* the redness (+) or greenness (−), and b* the yellowness (+) or blueness (−). The color was measured on the surface of 2 portions of lettuce for each replicate. The instrument was calibrated with a white plate as standard reference (L* = 97.55, a* = 1.32, b* = 1.41). The a* and b* color parameters recorded were used for the calculation of the hue angle (h°) using the formula: h° = arctg(b*/a*).

Organoleptic tests as subjective measurements and visual images as objective analyses related to product quality were carried out for trials 1 and 2 performed in Spain due to the absence of the same equipment in the collaborating laboratory in Denmark.

### 2.4. Statistical Analysis

The microbial data generated were log10-transformed and analyzed using a non-parametric test. Based on the nature of the experiments and the final adjustment of the data, the selected approach was a mixed model. *p* values below 0.05 were considered statistically significant. A Shapiro–Wilk test was performed to assess the normality of the microbiological data (*p* > 0.05). A Kruskal–Wallis test was used to determine differences between treatments for microbiological and organoleptic data during shelf life. The statistical analysis was performed using the R Statistical Software (v4.3.1; R Core Team 2021) as a language and environment for statistical computing (R Foundation for Statistical Computing, Vienna, Austria. URL http://www.R-project.org/; accessed on: 20 July 2023).

## 3. Results and Discussion

### 3.1. Application Challenges in the Industrial Settings

In this validation study, three main aspects were considered critical when bacteriophages were applied as post-process treatment in fresh produce: (1) the final concentration of the bacteriophages being applied, (2) the amount of water added to the final product and its quality impact, and (3) the uniformity application of the bacteriophage on the produce surface [21].

The application of the bacteriophage solution in the industrial setting required a preliminary optimization setup for the processing line characteristics such as: the speed of the conveyor belt, the amount and height of the washed product placed on the conveyor belt, and the pressure of the nozzles system. As a result of these tests, it was concluded that the treatment should be applied to the product at a flow rate of 3.33 mL/s and a minimum pressure of the device of 7 bar with a spray pattern of full-cone nozzle [22].

The prototype was placed above the vibration conveyor belt for the homogeneity distribution of the product and the proper application of the phage solution. The post-process treatment was applied for approximately 15 min. From the whole production, a total of 40 bags (500 g) were collected (20 bags of treated and another 20 bags of untreated product). Bags were transported to the lab for further analysis. After the treatment was applied, the processing lines were washed with water to eliminate any phage residue. Two trials were performed in each industrial setting. Some adjustments related to height and nozzle distribution were made after the first test. For the application point selection, the recommendations obtained by Leverentz et al. (2004) [23] that applied a commercial phage cocktail (phage mixture LMP-102, Listshield™, Intralytix, Inc., Baltimore, Maryland, US) with a spray gun were considered. These authors studied if the timing had an impact on phage effectiveness against Lm. They concluded that on fresh-cut melon, phage application was most effective between 0 and 1 h before contamination with Lm. This fact suggests that application should be carried out before packaging to be effective against potential contamination occurred during processing. In a previous work [11], we studied, at lab scale, two points of application of PhageGuard Listex™: the conveyor belt and the centrifuge. We observed that industrial application always poses challenges and encountered in curly endive that initial levels of Lm were reduced without significant differences among the point of application. In agreement with us, recently, Lu et al. (2022) [24] confirmed that inlet water used for phage dilutions should be free of chlorine residue to avoid phage reduction due to antimicrobial activity. These authors also recommended that sufficient rinsing should take place after the use of sanitizers in the washing operation before phages are applied.

### 3.2. Achievement of Phage Concentration and Listeria monocytogenes Presence

Based on the manufacturer’s recommendations, and also in our previous research, the target concentration of bacteriophages in the product was established at a level of 10^6^–10^7^ pfu/g [11]. Figure 2 shows the changes in the concentration of the bacteriophages obtained in the four sets of trials and during the storage. While the initial concentration of bacteriophages in the product of three out of the four trials was within the selected range, in the first trial, a lower concentration in the treated product was reached. The lack of device adaptation in trial 1 was solved by the mechanical adjustments of the nozzle and the prototype height. The changes made in the application device over the conveyor belt helped to reach the desired concentration. Between the two trials conducted in Spain, the concentration of phages declined more rapidly in trial 2 than in trial 1. One of the reasons could be the high CO_2_ concentration generated in the bags as a consequence of the anaerobic respiration probably due to the product quality changes, achieving at day 5 a 25% CO_2_ in trial 2 versus 16% CO_2_ in trial 1. A recent paper on the use of modified atmosphere packaging (MAP) combined with lactic acid bacteria (LAB) as bioprotective agents in cooked meat products showed that phages were not affected by concentrations of 20% CO_2_ [25]. However, in our study, higher CO_2_ concentrations were reached.

Another aspect was the processing conditions and, in particular, the application including the amount of product passing through the line that could vary and affect the phage–product interaction. We observed that within a range between 5 and 7 log of phages per gram of product, there were no detrimental effects on the sensorial characteristics when compared with the untreated product. The concentration of phages was always higher in trials 3 and 4 that were performed in Denmark because, even though they used the same application device, the processing line was smaller (approx. 500 kg/h) instead of the bigger processing amount in the line in the Spanish company (approx. 1000 kg/h).

Nebulization has been demonstrated to have destructive effects on phage structural stability [26]. These authors compared three nebulizers that were able to apply a titer dose of bacteriophages on aerosol of about 10^8^–10^9^ pfu/mL and a loading dose on the product of 10^10^ pfu/g. In the present experiment, the loss observed in the solution coming out of the nozzles was less than 1 log unit.

Another important aspect was the stability of the phages after application and during storage of the product. In these experiments, bacteriophage levels were determined at 0, 1, 5, and 9 days of shelf-life. As Test 2 lasted 12 days because the product was not spoiled, the product was examined until this day, although for microbiological analyses, only 9 days were considered. Table 1 shows the results of Kruskal–Wallis tests comparing bacteriophage counts in produce among days of storage in each test. In tests 1, 3, and 4, no significant differences were found between bacteriophage counts through storage. In these tests, the differences in phage log between the beginning and the end of storage ranged from 0.53 to 1.38 log pfu/g, in agreement with Guenter et al. (2008) [27]. These authors reported a reduction in bacteriophage AP511 in iceberg lettuce and cabbage stored at 6 °C for 6 days from 0.6 to 1.2 logs. The same authors reported a decrease in infective particles up to 2 logs when produce was stored at 20 °C. Many variables should be controlled in experiments at an industrial level compared to experiments at a laboratory level, especially when biocontrol treatments are involved [28].

Significant differences were found among tests in the phage titer of the produce ran in different places (Kruskal–Wallis chi-squared = 25.842, df = 3, *p*-value = 1.029 × 10^−5^). These differences may be due to the variation in the scale operations in the two processing plants [29]. The design of the processing lines was similar but not the same, and this could affect the efficiency of the prototype nebulizer, as it was designed for one factory and adapted for the other.

The efficacy of the application of the bacteriophage treatment in commercial products can only be confirmed if Lm is naturally present. In the four trials performed in the industrial settings, colonies compatible with *Listeria* spp. were found up to 2.00 log. These microorganisms could be present in the raw material entering the processing plants (RTE and frozen produce industry) [30]. While Lm contamination is a potential risk, *Listeria* spp. could serve as an indicator microorganism indicating the possible entrance of Lm in industrial settings [3]. However, none of the presumptive *Listeria* spp. were confirmed as Lm. Data obtained under lab-scale trials showed that at the level of phages achieved in iceberg lettuce, Lm log reductions were confirmed, up to 3.0 log CFU/g after 15 days of storage. Guenter et al. (2009) [27] registered a decrease in more than 2 log units in two Lm strains in iceberg lettuce leaves treated with the phage A511. In the same study, a reduction in Lm counts up to 2 log units was achieved in cabbage treated with phage P100 (initial Lm inoculum level: 3.00 log cfu/g) [27]. In both cases, an increase in the Lm population due to the multiplication of the surviving bacteria cells was registered. Truchado et al. (2020) [11] reported a decrease in Lm counts in fresh-cut curly endive of an average of about 2.5 logs but registered an increase in bacterial counts after 8 days of storage (3 days 5 °C + 5 days 8 °C). On the other hand, Leverentz et al. (2004) [23] achieved a reduction in Lm up to 6.8 log units after 7 days of storage of honeydew melons pieces at lab scale with another commercial phage cocktail (phage mixture LMP-102, Listshield™, Intralytix, Inc.). Similar findings were reported by Lone et al. (2016) [31] in fresh-cut cantaloupe inoculated with Lm. These authors registered a decrease in pathogen counts higher than 2 logs when treated with a phage cocktail (LinM-AG8, LmoM-AG13, and LmoM-AG20) and stored at 4 and 12 °C.

The type of commodity can influence the efficacy of the treatment. Thus, Oliveira et al. (2014) [32] observed that higher pH and liquid formats (fruit juices) influence positively treatment results. Leverentz et al. (2003) [12] found that phage mixtures LM-10^3^ and LMP-10^2^ were effective against Lm in honeydew melon but not in Red Delicious slices. Other authors have observed the same tendency and explained these findings due to the pH of the fresh product [32]. An interaction between phages and their hosts can be influenced by other factors apart from pH and food matrix. Some authors have highlighted that ionic strength, the presence of substances that interfere with phage particle diffusion or penetration through the cell wall and membrane among other parameters, are factors that influence interactions between the virus and the host [33]. Some bioactive compounds present in fresh produce can also have a negative effect on bacteriophage titer [34]. Iceberg lettuce has a pH of approximately 6 [35], which, in principle, is not expected to produce a negative effect on the bioagent mechanism. In our study, validation was performed following the commercial operations conducted by the industry, which means that bacteriophages were applied to washed in chlorinated water, and then rinsed iceberg lettuce. This fact could reduce some natural compounds present in fresh produce such as ascorbic acid [36], whose oxidation products have been reported to be inactive in some bacteriophage strains [37].

### 3.3. Sensory Analyses

A critical challenge to consider in the phage application is the amount of water used for the liquid solution. In our previous studies used as a guide for the present trials, a lab-scale spray system (Spraying Systems CO^®^ device AUTOJET model 1550+) with a lab scale tank and a J series straight nozzle was used for treatment application. This device managed to reduce the amount of water applied to the product (0.3 mL/s) as well as reducing aerosol formation avoiding the possible dispersion of the treatment solution in the working environment. In the present study, a spraying system described in Material and Methods section was used with a higher flow of water (3.33 mL/s), which resulted in a greater addition of water to the product in comparison to lab scale described. In theory, this might be thought to accelerate deterioration processes in treated product compared to control [38,39], but no significant differences in the organoleptic evaluation were found between treated and untreated product.

Before opening the bags, headspace gas composition was measured including O_2_ and CO_2_ levels. The concentration of CO_2_ in both trials increased to approximately 30% while O_2_ concentration decreased to 0% approximately. No significant differences were found between treated and untreated samples or between trials (Table 2). The anaerobic conditions reached were due to the high respiration rate of shredded lettuce because of the high wound response that accelerated the metabolism and increased the respiration rate to pieces of higher size [40]. The temperature of storage also affected the increase in the respiration rate and consequence the modified atmosphere packaging, increasing CO_2_ levels as a consequence of the anaerobic metabolism, that reduced the shelf life. Similar tendencies have been observed by other authors as the expected evolution of this packaging conditions used [40].

When color changes were studied by hue angles and L* values, no significant differences were found between the color parameters of the product treated with phages and control samples in test 1 (Table 2, Figure 3 and Figure 4). Significant differences between treated and untreated lettuce in test 2 at day 0 were found regarding these two color parameters and these differences were maintained over the storage. These differences were not referable to the treatments but to the color differences between batches. These differences were captured by the objective measure of image analysis that was able to separate batches of and detected color differences, as previously reported in vegetables with different degrees of green color [41].

Similarly, no differences were found among the organoleptic parameters measured by the sensory panelists when comparing treated and untreated shredded lettuce (Table 2). Sensory panel perception scored similarly in the case of untreated samples and samples treated with phages. Sensory results agree with previously reported results by the processors and expert panel regarding the impact of phages on the quality of the product [41].

## 4. Conclusions

Our results allowed us to optimize the utilization of PhagueGuard Listex™ as a post-process treatment counteracting the main difficulties of phage application at an industrial level. Particularly, we optimized the method of application that included the device and the process operation steps and achieved the application of the proper concentration of phages by a fine, mist-like spray with no phage inactivation, and the adequate coverage of the product surface. Future research is still ongoing about the fate of the possible persistence of phages once applied in an industrial environment, which is one important aspect for optimal application.

## Figures and Tables

**Figure 1 foods-12-03171-f001:**
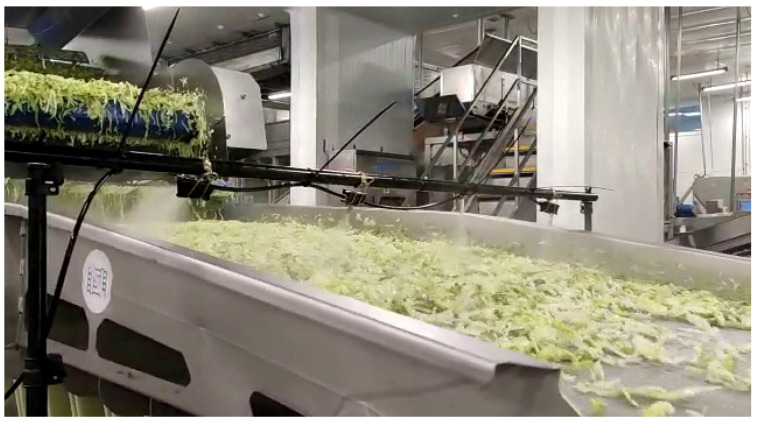
Conveyor belt selected as the point of application and the prototype installed above it for the phage solution application as a fine, mist-like spray covering uniform the entire product surface.

**Figure 2 foods-12-03171-f002:**
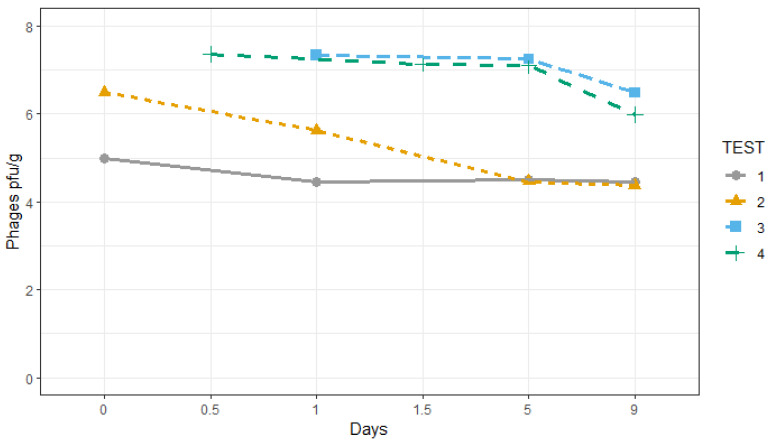
Changes in the bacteriophage concentration applied in the 4 trials conducted in industrial settings in shredded iceberg lettuce stored for 9 days at 7 °C.

**Figure 3 foods-12-03171-f003:**
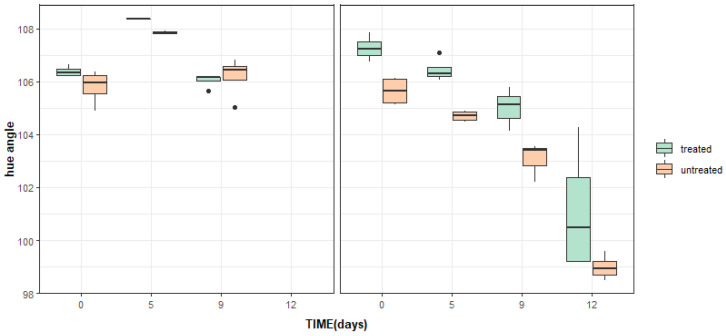
Changes in hue angle between untreated and post-process Listex™-treated iceberg lettuce over 9 days of storage at 7 °C in test 1 (**left graph**) and test 2 (**right graph**). Box plots represent the interquartile interval, where 50% is the median (middle quartile) (n = 4 per day and treatment) and the lower and upper quartiles are 25 and 75% of the scores, respectively. Symbols (●) represent outliers.

**Figure 4 foods-12-03171-f004:**
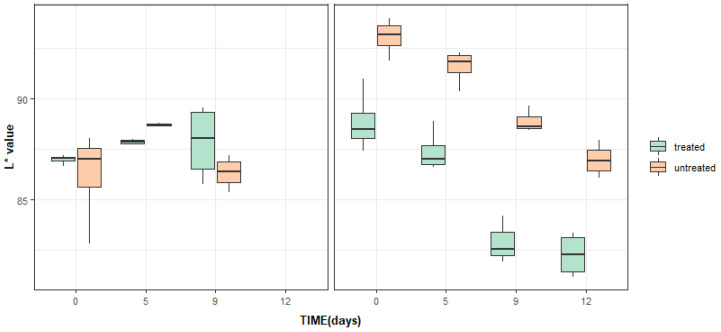
Changes in L* value between untreated and post-process Listex™-treated iceberg lettuce over 9 days of storage at 7 °C in test 1 (**left graph**) and test 2 (**right graph**). Box plots represent the interquartile interval, where 50% is the median (middle quartile) (n = 4 per day and treatment) and the lower and upper quartiles are 25 and 75% of the scores, respectively.

**Table 1 foods-12-03171-t001:** Kruskal–Wallis test comparison in each test of logs phages pfu/g of treated samples through 9 days of storage at 7 °C.

TESTS	K-W Test
TEST 1	Chi-square 6.633
	df 3
	*p* value 0.084
TEST 2	Chi-square 12.654
	df 4
	*p* value 0.0131
TEST 3	Chi-square 3.429
	df 2
	*p* value 0.180
TEST 4	Chi-square 6.167
	df 3
	*p* value 0.103

**Table 2 foods-12-03171-t002:** Kruskal–Wallis test comparison between headspace gas composition (O_2_ and CO_2_), color parameters (hue angle and L*) and organoleptic parameters between treated and untreated samples through 9 days (test 1) and 12 days (test 2) of storage at 7 °C.

Quality Parameters	TEST 1			TEST 2		
	Chi-Square	dF	*p* Value	Chi-Square	dF	*p* Value
CO_2_	0.404	1	0.525	3.411	1	0.065
O_2_	0.270	1	0.603	0.460	1	0.498
L*	6.313	1	0.0119	10.874	1	0.000
Hue angle	4.597	1	0.032	4.924	1	0.026
Flavor	0.009	1	0.922	0.000	1	1
Texture	0.001	1	0.974	0.000	1	0.987
Browning	0.003	1	0.958	0.031	1	0.820
Spoilage	0.018	1	0.895	0.028	1	0.868
Visual appearance	0.031	1	0.861	0.031	1	0.860

## Data Availability

Data is contained within the article.

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
