# Peer review of "Optimization of the Use of a Commercial Phage-Based Product as a Control Strategy of Listeria monocytogenes in the Fresh-Cut Industry"

_foods, 2023, doi:10.3390/foods12173171_

Round 1

Reviewer 1 Report

Indeed the manuscript is well written. The narrative in each section is easily followed. Yet my main concern is the discrepancy between the title and project description. As I read the title my impression was that the methodology was going to be focused on controlling LM inoculated on fresh-cut. Then, I read the abstract and noticed that initial and final counts of listeria and phages were missing. So, I moved to the methodology section and confirmed that no challenge trials were conducted in fresh produce industry lines.  I suggest that the manuscript should be redirected to the effect of phage inoculation on the sensory quality of Lettuce.

Author Response

We are sorry that by the title the reviewer understood that the objective of our study was the evaluation through challenge tests of the potential of bacteriophages as biopreservative agents. The aim of our study, however, was the optimization of their mode of application in two fresh-cut factories for the control of Listeria monocytogenes in ready-to-eat products and the effect of phage inoculation on the sensory quality of fresh-cut lettuce.

Considering the other reviewers, the title has been changed as well as the description of the study, making clearer the focus on the optimization of the application and the effect on the sensory characteristics following your suggestions. The term ‘challenges’ was used as a synonym to overcome the difficulties of the industrial application of a commercial biopreservation agent. Now, the title, abstract, and material and method sections have no discrepancies. The counts of Listeria and phages have been described in more detail.

Reviewer 2 Report

Foods Manuscript 2550471

“Industrial validation challenges of bacteriophages as a control strategy of Listeria monocytogenes in the fresh-cut industry”

General comments

The manuscript deals with the optimization at an industrial level of the use of a well known phage-based commercial product (PhageGuard ListexTM) to control Listeria monocytogenes in the industrial production cycle of shredded iceberg lettuce. A challenge test with the pathogenic microorganism was not carried out since it was not the aim of the study. The aim of the study was to define the correct method of application using a prototype device to obtain the correct level of phages concentration in the product and to verify the impact on the quality of the final product. The topic is of interest as it aims to bridge the gap between laboratory and field data.

Some issues have to be addressed, as specified in “Specific comments”.

Specific comments

Title: it does not fully reflect the manuscript content. It suggests that a microbial challenge test was performed. To be changed. A suggestion: Optimization of the use of a commercial phage-based product as a control tool of Listeria monocytogenes in the fresh-cut industry, or something similar.

Introduction, line 29: not “bacteria” but “bacterium” or “microorganism”.

Introduction, line 30: Add a more specific reference about the first sentence of Introduction.

Introduction, line 30: not “their detection” but “Lm detection”.

Introduction, lines 49-50: “although very little in fresh whole products and almost unknown in RTE products”: not clear, maybe “in RTE products of vegetal origin? To be considered that meat and dairy products such as sausages and cheeses are RTE…

Introduction, lines 63-64: The sentence “This study aimed to determine the suitability of a commercial phage-based treatment to control Lm in leafy greens at an industrial scale.” has to be reformulated. To determine the suitability, the Authors should have performed a microbial challenge test with Listeria monocytogenes in the tested conditions. Better “to define method and parameters of application of a commercial phage-based treatment to control Lm in leafy greens at an industrial scale” or similar.

Materials and methods 2.1, line 81: not “Validation of PhageGuard Listex™ in an industrial setting” but “Optimization of the utilisation of PhageGuard Listex™ in an industrial setting”

Materials and methods 2.2: the Authors did not carry out an enumeration but a detection of Listeria monocytogenes since the applied ISO 11290-1 with slight modifications is a detection method (qualitative method).

Materials and methods 2.2: please, add a reference for the used PCR.

Materials and methods 2.3: “Organoleptic tests as subjective measurements and visual images as objective analyses related to product quality were carried out in Spain due to the absence of the same equipment in the collaborating laboratory in Denmark.” Please, specify more clearly: were organoleptic tests and visual images carried out only for trial 1 and 2 performed in Spain? Moreover, it is better to move this sentence at the end of 2.3, after the analyses have been described.

Results and discussion 3.1, line 166: colon instead of full stop before 1).

Results and discussion 3.1, line 172: not “high” but “height”.

Results and discussion 3.1, line 179: 40 treated bags and 40 control bags? To be written more clearly.

Results and discussion 3.1, line 187: “They concluded that on this type of product…” what type?

Results and discussion 3.1, line 190: “Truchado et al. (2020) tested…”: better “we studied in a previous work…”

Results and discussion 3.2: the title “Antimicrobial effect” is not appropriate since results are not about antimicrobial (anti-Lm) activity but about phages concentration. To be changed.

Figure 2: why the starting point for test 3 and 4 is 0.5 days instead of 0 days as for test 1 and 2?

Results and discussion 3.2 line 215: what is the outlet solution? The sentence is not clear, to be rewritten.

Results and discussion 3.2 lines 219-221: in test 2 phages concentration decreased to less than 6 pfu/g after 1, 5 and 9 days. Can you discuss this result?

Table 1: not “Kruskal- Wallis test comparison between tests of logs phages pfu/g of treated samples through 9 days of storage at 7ºC.” but “Kruskal- Wallis test comparison in each test of logs phages pfu/g of treated samples through 9 days of storage at 7ºC”

Results and discussion 3.2 lines 240-241: “In the four trials performed in the industrial settings, colonies compatible with Listeria spp. were found up to 2.00 log”. How can the Authors state “up to 2 log” considering that they carried out a qualitative determination with an enrichment phase (presence/absence in 25 g)? The Authors can only state that colonies compatible with Listeria spp. were found.

Results and discussion 3.2 line 262: “…higher pH and liquid formats…” Not clear what “liquid format” means. Maybe level of activity water or water content?

Results and discussion 3.2 lines 288-290: move the references 34-35 after “In theory, this might be thought to accelerate deterioration processes in treated product compared to control”.

Results and discussion 3.3 lines 288-290: the sentence “Significant differences between treated and untreated lettuce in test 2 at day 0 were found regarding these two color parameters and these differences were maintained over the storage” referred to hue and L* is not clear, considering that in Table 2 the P value of L* is 0.820 in test 2.

Conclusions:

line 330: it is not correct to state “Our results show that the validation of PhagueGuard Listex™ as a post-process treatment was a relevant achievement” since the study was not focused on the validation of PhagueGuard Listex™, but of its utilisation at industrial level. Please, reformulate the sentence.

Analogously, at the point iii) it is not correct to state that “the adequate coverage of the product surface and thus the effective effect of biocontrol of Lm log reductions achieved “but only that “ the adequate coverage of the product surface was achieved”.

line 332: add a colon before i)

References: The Authors cited as a references their paper “Gómez-Galindo, M.; Truchado, P.; Volpi, M.; Elsser-Gravesen, A.; Gil1, M.I.; Allende, A. Inactivation Efficacy of 405 Four Commercial Bioprotective Post-Process Treatments against Listeria Monocytogenes and Impact on the 406 Commercial Quality of Leafy Greens. Food Control 2023.(submitted). Since it is at present submitted but not yet accepted, it is not publicly available. Thus, it is not acceptable as a reference. Modify the text, accordingly (line 280).

Author Response

REVIEWER 2

General comments

The manuscript deals with the optimization at an industrial level of the use of a well-known phage-based commercial product (PhageGuard ListexTM) to control Listeria monocytogenes in the industrial production cycle of shredded iceberg lettuce. A challenge test with the pathogenic microorganism was not carried out since it was not the aim of the study. The aim of the study was to define the correct method of application using a prototype device to obtain the correct level of phages concentration in the product and to verify the impact on the quality of the final product. The topic is of interest as it aims to bridge the gap between laboratory and field data.

Thank you very much for your comments and the correct description of our manuscript objectives and results.

Some issues have to be addressed, as specified in “Specific comments”.

Specific comments

Title: it does not fully reflect the manuscript content. It suggests that a microbial challenge test was performed. To be changed. A suggestion: Optimization of the use of a commercial phage-based product as a control tool of Listeria monocytogenes in the fresh-cut industry, or something similar. The title has been changed according to the reviewer’s recommendations.

Introduction, line 29: not “bacteria” but “bacterium” or “microorganism”. Corrected

Introduction, line 30: Add a more specific reference about the first sentence of Introduction. A more specific reference has been included.

Introduction, line 30: not “their detection” but “Lm detection”. This part of the sentence has been deleted according to the reviewer’s 3 suggestions.

Introduction, lines 49-50: “although very little in fresh whole products and almost unknown in RTE products”: not clear, maybe “in RTE products of vegetal origin? To be considered that meat and dairy products such as sausages and cheeses are RTE… It has been changed as suggested.

Introduction, lines 63-64: The sentence “This study aimed to determine the suitability of a commercial phage-based treatment to control Lm in leafy greens at an industrial scale.” has to be reformulated. To determine the suitability, the Authors should have performed a microbial challenge test with Listeria monocytogenes in the tested conditions. Better “to define method and parameters of application of a commercial phage-based treatment to control Lm in leafy greens at an industrial scale” or similar. It has been changed as suggested.

Materials and methods 2.1, line 81: not “Validation of PhageGuard Listex™ in an industrial setting” but “Optimization of the utilisation of PhageGuard Listex™ in an industrial setting”. It has been changed as suggested.

Materials and methods 2.2: the Authors did not carry out an enumeration but a detection of Listeria monocytogenes since the applied ISO 11290-1 with slight modifications is a detection method (qualitative method). The enumeration of Lm after filtration was also conducted although L. monocytogenes was not detected in any of the four tests performed. The information on the enumeration protocol carried out is now included in the text asDetection and enumeration of Listeria spp. and Lm in iceberg lettuce were based on the ISO-11290-1 and ISO-11290-2 methods with slight modifications.

Materials and methods 2.2: please, add a reference for the used PCR. The PCR reference used has been mentioned.

Materials and methods 2.3: “Organoleptic tests as subjective measurements and visual images as objective analyses related to product quality were carried out in Spain due to the absence of the same equipment in the collaborating laboratory in Denmark.” Please, specify more clearly: were organoleptic tests and visual images carried out only for trial 1 and 2 performed in Spain? Moreover, it is better to move this sentence at the end of 2.3, after the analyses have been described. Corrected.

Results and discussion 3.1, line 166: colon instead of full stop before 1). Ok

Results and discussion 3.1, line 172: not “high” but “height”. Ok

Results and discussion 3.1, line 179: 40 treated bags and 40 control bags? To be written more clearly. This sentence has been changed to ‘……a total of 40 bags (500 g) was collected (20 bags of treated and another 20 bags of untreated product)’.

Results and discussion 3.1, line 187: “They concluded that on this type of product…” what type? Corrected

Results and discussion 3.1, line 190: “Truchado et al. (2020) tested…”: better “we studied in a previous work…”. Corrected.

Results and discussion 3.2: the title “Antimicrobial effect” is not appropriate since results are not about antimicrobial (anti-Lm) activity but about phages concentration. To be changed. The title has been changed as ‘Achievement of phage concentration’.

‘.Figure 2: why the starting point for test 3 and 4 is 0.5 days instead of 0 days as for test 1 and 2? Due to the distance between the processing facility and the lab in Denmark, the analyses were conducted 12 h after processing while in Spain, the distance was very short and the analyses were done in less than 1 h after processing.

Results and discussion 3.2 line 215: what is the outlet solution? The sentence is not clear, to be rewritten. The sentence has been clarified as ‘ …coming out of the nozzles..

Results and discussion 3.2 lines 219-221: in test 2 phages concentration decreased to less than 6 pfu/g after 1, 5 and 9 days. Can you discuss this result? A detailed discussion on this important issue has been included. In agreement with the reviewer this is an important aspect for understanding the variability in the industrial application of this treatment. In our previous study, we observed the favorable antilisterial activity of this commercial bacteriophage agent achieving the inhibition of at least 2 log the pathogen inoculated on different fresh-cut products, including cut lettuce. In the present study, there were some aspects related to the product and the processing that could affect the phage-product interaction. One was the quality of the product. We performed 4 trials as the quality of the product could varied as thus affect the impact of phage-product interaction. Another aspect was the processing conditions in particular the application such as the amount of product passing through the line could varied and may affect the phage-product interaction. We observed that within a range between 5-7 log of phages per gram of product, there were no detrimental effects on the sensorial characteristics when compared with the untreated product. The concentration of phages was always higher in trials 3 and 4 performed in Denmark because even though they used the same device, the processing line was smaller (approx. 500 kg/h) instead of the bigger processing amount in the line (1000 kg/h) in the Spanish company. Between the two trials conducted in Spain the concentration declined rapidly in trial 2 than in trial 1. On of the reasons could be the high CO2 concentration generated in the bags as a consequence of the anaerobic respiration probably due to quality changes, achieving at day 5 a 25% CO2 in trial 2 versus 16% CO2 in trial 1. A recent paper on the use of modified atmosphere packaging (MAP) (20 % CO2–80 % N2) combined with lactic acid bacteria (LAB) as bioprotective agents showed no reductions on LAB in cooked meat products (Barcenilla et al., 2023).

Table 1: not “Kruskal- Wallis test comparison between tests of logs phages pfu/g of treated samples through 9 days of storage at 7ºC.” but “Kruskal- Wallis test comparison in each test of logs phages pfu/g of treated samples through 9 days of storage at 7ºC” Corrected

Results and discussion 3.2 lines 240-241: “In the four trials performed in the industrial settings, colonies compatible with Listeria spp. were found up to 2.00 log”. How can the Authors state “up to 2 log” considering that they carried out a qualitative determination with an enrichment phase (presence/absence in 25 g)? The Authors can only state that colonies compatible with Listeria spp. were found. As stated before, enumeration was performed, thus this sentence has not been modified.

Results and discussion 3.2 line 262: “…higher pH and liquid formats…” Not clear what “liquid format” means. Maybe level of activity water or water content? This sentence has been clarified according to the reference as’..liquid formats (fruit juices)..

Results and discussion 3.2 lines 288-290: move the references 34-35 after “In theory, this might be thought to accelerate deterioration processes in treated product compared to control”. Corrected.

Results and discussion 3.3 lines 288-290: the sentence “Significant differences between treated and untreated lettuce in test 2 at day 0 were found regarding these two color parameters and these differences were maintained over the storage” referred to hue and L* is not clear, considering that in Table 2 the P value of L* is 0.820 in test 2. Corrected. The value corresponds to a sensory panel characteristic measured; when reordering the table, the data were crossed.

Conclusions:

line 330: it is not correct to state “Our results show that the validation of PhagueGuard Listex™ as a post-process treatment was a relevant achievement” since the study was not focused on the validation of PhagueGuard Listex™, but of its utilisation at industrial level. Please, reformulate the sentence. This sentence has been modified as ‘Our results show the utilization of PhagueGuard Listex™ as a post-process treatment that counteracts the main difficulties of phage application at an industrial level’.

Analogously, at the point iii) it is not correct to state that “the adequate coverage of the product surface and thus the effective effect of biocontrol of Lm log reductions achieved “but only that “ the adequate coverage of the product surface was achieved”. Corrected

line 332: add a colon before i). Ok

References: The Authors cited as a references their paper “Gómez-Galindo, M.; Truchado, P.; Volpi, M.; Elsser-Gravesen, A.; Gil1, M.I.; Allende, A. Inactivation Efficacy of 405 Four Commercial Bioprotective Post-Process Treatments against Listeria Monocytogenes and Impact on the 406 Commercial Quality of Leafy Greens. Food Control 2023.(submitted). Since it is at present submitted but not yet accepted, it is not publicly available. Thus, it is not acceptable as a reference. Modify the text, accordingly (line 280). The reference has been deleted and a mention of ‘own results or in our previous studies’ has been included in the text.

Reviewer 3 Report

Abstract:

Line 15: change "selected" to "using"

Line 16: change to "(RTE), high process volume product."

Line 19: change " the correct" to "proper"

Introduction:

Line 29: change "cause" to "causes"

Line 30-31: delete "and their detection....product recalls"

Line 33: delete "records related to Lm"

Line 37: delete "and specifically Lm"

Line 38: Delete "Fortunately" and start with "The risk of ..."

Line 54: change "eclipsed" to more appropriate word...such as "overshadowed"? 

Line 56: what are "consumer concerns"?

Lines 65-66: re-write as not clear what goal to evidence is being described?

Line 68: "active microorganisms" is misleading as phages are not metabolically active per se, and require a metabolically active bacteria in order to have the intended effect.

Material and Methods

Line 74: is it only P100? Why do authors write "a cocktail of phages"?

Line 97: For Figure 1, it would be nice to see the actual product being sprayed? What about the underside (i.e., what is thickness of product as it passes) and phage application?

Line 113: change to "Broth and 1% pyruvate..."

Line 114: change to "Then, 1 ml..."

Line 115: missing information on what agar(s) were used as PCR described on colonies. As well, reference for PCR method? If in-house then need conditions.

Results and Discussion:

Line 165: delete "the" and replace with "this"

Line 166: change to "the final concentration of the bacteriophages being applied, 2)..."

Line 172: change "high" to "height" if that is what authors mean?

Line 178: what does "passed" mean? Is it drying after application?

Line 206: "regulation" - is this control of?

Figure 2: what changed from trial 1 to trial 2 as there is a large difference to trials 3 and 4 (3 and 4 are very consistent and similar which I assume is desired outcome)?

Lines 217-226: how were phages measured? What is protocol? This is critical and no information given

Line 244: change "index" to "indicator"

Lines 246-248: should actually present data in a supplementary section or directly in manuscript as there no Listeria species isolated during this work so it is not possible to know if the application actually worked?

Line 251: "regrowth" does this refer to cells infected by the phages? It is almost implying that the phage treatment does not really work other than slowing growth temporarily? or Does it actually kill some?

Line 272: change "of around" to "approximately"

Lines 286-290: what is the water activity differences between treated and untreated samples? Are treated samples left to dry prior to packaging? 

Figures 3 and 4: It is not clear why information is shown for test 1 and test 2 that were inconsistent? Authors should present the data for tests 3 and 4 where optimal levels of phages were shown.

Conclusions

Line 330: it is not clear what actually was validated? There needs to be information as to how phage titres were actually assessed

Line 331: what is "relevant achievement"? 

Line 334: "no phage inactivation" - again, how was this tested?

Line 335: no data is presented t show "effect effect of control" so this is misleading and incorrect.

Comments and editorial suggestions given in comments to authors above.

Author Response

REVIEWER 3

Abstract:

Line 15: change "selected" to "using". Corrected.

Line 16: change to "(RTE), high process volume product." Corrected.

Line 19: change " the correct" to "proper". Corrected.

Introduction:

Line 29: change "cause" to "causes". Corrected.

Line 30-31: delete "and their detection....product recalls". Corrected.

Line 33: delete "records related to Lm". Corrected.

Line 37: delete "and specifically Lm". Corrected.

Line 38: Delete "Fortunately" and start with "The risk of ...". Corrected.

Line 54: change "eclipsed" to more appropriate word...such as "overshadowed"? Corrected.

Line 56: what are "consumer concerns"? This sentence has been changed toIn some countries, the use of different phages as biocontrol agents has been approved [14], although there are still some concerns regarding to their approval as processing aids.’

Lines 65-66: re-write as not clear what goal to evidence is being described? This sentence has been changed according to the reviewer’s 2 suggestions.

Line 68: "active microorganisms" is misleading as phages are not metabolically active per se, and require a metabolically active bacteria in order to have the intended effect. Corrected

Material and Methods

Line 74: is it only P100? Why do authors write "a cocktail of phages"? Corrected

Line 97: For Figure 1, it would be nice to see the actual product being sprayed? What about the underside (i.e., what is thickness of product as it passes) and phage application? The photo has been changed to another one where product is clearly seen how it was sprayed. The thickness of the product is approximately (few lettuce leaves strips) and the conveyor belt vibrates continuously so the product is well mixed.

Line 113: change to "Broth and 1% pyruvate...". Corrected.

Line 114: change to "Then, 1 ml...". Corrected.

Line 115: missing information on what agar(s) were used as PCR described on colonies. As well, reference for PCR method? If in-house then need conditions. The information has been included.

Results and Discussion:

Line 165: delete "the" and replace with "this". Corrected.

Line 166: change to "the final concentration of the bacteriophages being applied, 2)...". Corrected

Line 172: change "high" to "height" if that is what authors mean?. Corrected.

Line 178: what does "passed" mean? Is it drying after application? This part has been deleted.

Line 206: "regulation" - is this control of? This sentence has been changed toThe lack of the device adaptation’.

Figure 2: what changed from trial 1 to trial 2 as there is a large difference to trials 3 and 4 (3 and 4 are very consistent and similar which I assume is desired outcome)? As previously mentioned, there were some differences between the trials 1 and 2 conducted in Spain and the trials 3 and 4 conducted in Denmark. The main one was the processing capacity per hour which mean amount of product that passes per hour. In the Spanish processing facility, the production yield was double than that of the Denmark processing line.

Lines 217-226: how were phages measured? What is protocol? This is critical and no information given. A brief description and a reference have been included.

Line 244: change "index" to "indicator". Corrected.

Lines 246-248: should actually present data in a supplementary section or directly in manuscript as there no Listeria species isolated during this work so it is not possible to know if the application actually worked? As mentioned before, the antilisteria effect was previously observed in a study conducted in lab conditions in which the cut lettuce was inoculated with 3-4 log of Lm before washing

Line 251: "regrowth" does this refer to cells infected by the phages? It is almost implying that the phage treatment does not really work other than slowing growth temporarily? or Does it actually kill some? PhageGuard Listex has been successfully proven to kill Listeria (commercial tests and our previous work have demonstrated this aspect). However, Listeria can regrowth if any cell remains alive and begins to multiply.

Line 272: change "of around" to "approximately". Corrected.

Lines 286-290: what is the water activity differences between treated and untreated samples? Are treated samples left to dry prior to packaging?  No, both treated and untreated products were processed in the same way except for the phage application.

Figures 3 and 4: It is not clear why information is shown for test 1 and test 2 that were inconsistent? Authors should present the data for tests 3 and 4 where optimal levels of phages were shown. As mentioned in materials and methods (section 2.3), organoleptic tests as subjective measurements and visual images as objective anal-yses related to product quality were carried out for trials 1 and 2 performed in Spain due to the absence of the same equipment in the collaborating laboratory in Denmark.

Conclusions

Line 330: it is not clear what actually was validated? There needs to be information as to how phage titres were actually assessed. This sentence has been changed according to the reviewer’s 2 recommendations.

Line 331: what is "relevant achievement"? Modified.

Line 334: "no phage inactivation" - again, how was this tested? Explained already

Line 335: no data is presented t show "effect effect of control" so this is misleading and incorrect. It has been clarify based on the main conclusions.

Round 2

Reviewer 1 Report

No further comments

Author Response

Thank you very much for revising our manuscript and tell us that no new issues were detected

Reviewer 2 Report

Peer review 2

Foods Manuscript 2550471

General comments

The main issues have been addressed.

Please, revise english in the newly added parts as well as the reference list (Truchado et al., 2020 is repeated in 11 and 19). The same for °C and % (with or without space) and similar, to be checked.

Specific comments

Abstract line 19: was

Abstract line 23: The post-process treatment with PhageGuard ListexTM did…

Introduction line 30: not Listeria outbreaks  but Lm outbreaks

Introduction line 37: the risk of Lm final produce…

MM 2.2:  not a homogenized but a homogenate, not tampon but buffer, not were but was (line 119), not 19 but 11 (line 126).

Results 3.1 line 190: (1 sec last product below it): what does it mean?

Lines 204-206: We observed that industrial application always poses challenges and encountered in curly endive that initial levels of Lm were reduced without significant differences among the point of application. In agreement with us, recently Tong Lu et al.(2022) [25]…

Title 3.2: 3.2. Achievement of phage concentration and of Listeria monocytogenes presence, or similar.

3.2: “A recent paper on the use of modified atmosphere pack- aging (MAP) (20 % CO2–80 % N2) combined with lactic acid bacteria (LAB) as bioprotec- tive agents showed no reductions in LAB in cooked meat products [26]”. I do not understand the connection between this reference (used to show that LAB were not affected by CO2) and the previous sentence (phages were probably affected by the higher CO2 concentration)? Please clarify or delete.

Line 227: Another aspect was the processing conditions and in particular the application

228: not affected but affect

267-270: These mi- croorganisms could be present in the raw material entering the processing plants (RTE and frozen produce industry) [31]. While Lm contamination is a potential risk, Listeria spp. could serve as an indicator microorganism indicating the possible entrance of Lm in in- dustrial settings [3].

299: which, in principle, is not expected to produce a negative effect on the bioagent mechanism.

Table 2: of headspace…between treated and …

Conclusions: I suggest:

Our study allowed to optimize the utilization of PhagueGuard Listex™ as a post-process treatment counteracting the main difficulties of phage application at an industrial level. Particularly, we optimized the method of application that included the device and the process operation steps and achieved the application of the proper concentration of phages by a fine, mist-like spray with no phage inactivation, and the adequate coverage of the product surface.

or similar.

Author Response

General comments

The main issues have been addressed. Thank you very much for revising our manuscript. Your relevant questions and critical suggestions have improved the manuscript and corrected the mistakes.

Please, revise english in the newly added parts as well as the reference list (Truchado et al., 2020 is repeated in 11 and 19). The same for °C and % (with or without space) and similar, to be checked. The English in the new parts have been revised as well as the symbols ºC and %. The references have been checked and the repeated one by Truchado et al. (19) has been deleted.

Specific comments

Abstract line 19: was. Corrected

Abstract line 23: The post-process treatment with PhageGuard ListexTM did… Corrected

Introduction line 30: not Listeria outbreaks  but Lm outbreaks. Corrected

Introduction line 37: the risk of Lm final produce… Corrected

MM 2.2:  not a homogenized but a homogenate, not tampon but buffer, not were but was (line 119), not 19 but 11 (line 126). Corrected

Results 3.1 line 190: (1 sec last product below it): what does it mean? Deleted.

Lines 204-206: We observed that industrial application always poses challenges and encountered in curly endive that initial levels of Lm were reduced without significant differences among the point of application. In agreement with us, recently Tong Lu et al.(2022) [25]… Corrected

Title 3.2: 3.2. Achievement of phage concentration and of Listeria monocytogenes presence, or similar. Corrected

3.2: “A recent paper on the use of modified atmosphere pack- aging (MAP) (20 % CO2–80 % N2) combined with lactic acid bacteria (LAB) as bioprotec- tive agents showed no reductions in LAB in cooked meat products [26]”. I do not understand the connection between this reference (used to show that LAB were not affected by CO2) and the previous sentence (phages were probably affected by the higher CO2 concentration)? Please clarify or delete. This sentence has been clarified as ‘A recent paper on the use of modified atmosphere packaging (MAP) combined with lactic acid bacteria (LAB) as bioprotective agents in cooked meat products showed that phages were not affected by concentrations of 20% CO2 [26]. However, in our study, higher CO2 concentrations were reached.

Line 227: Another aspect was the processing conditions and in particular the application. Corrected

228: not affected but affect. Corrected

267-270: These mi- croorganisms could be present in the raw material entering the processing plants (RTE and frozen produce industry) [31]. While Lm contamination is a potential risk, Listeria spp. could serve as an indicator microorganism indicating the possible entrance of Lm in in- dustrial settings [3]. Corrected

299: which, in principle, is not expected to produce a negative effect on the bioagent mechanism. Corrected

Table 2: of headspace…between treated and … Corrected

Conclusions: I suggest:

Our study allowed to optimize the utilization of PhagueGuard Listex™ as a post-process treatment counteracting the main difficulties of phage application at an industrial level. Particularly, we optimized the method of application that included the device and the process operation steps and achieved the application of the proper concentration of phages by a fine, mist-like spray with no phage inactivation, and the adequate coverage of the product surface. Corrected as suggested.